# Resting Energy Expenditure, Body Composition and Phase Angle in Anorectic, Ballet Dancers and Constitutionally Lean Males

**DOI:** 10.3390/nu11030502

**Published:** 2019-02-27

**Authors:** Maurizio Marra, Rosa Sammarco, Emilia De Filippo, Carmela De Caprio, Enza Speranza, Franco Contaldo, Fabrizio Pasanisi

**Affiliations:** 1Department of Clinical Medicine and Surgery, University Federico II, 80131 Naples, Italy; marra@unina.it (M.M.); emiliotta@libero.it (E.D.F.); carmeladecaprio@iol.it (C.D.C.); enza.speranza@unina.it (E.S.); contaldo@unina.it (F.C.); pasanisi@unina.it (F.P.); 2Interuniversity Centre for Obesity and Eating Disorders (CISRODCA), Federico II University of Naples, 80131 Naples, Italy

**Keywords:** REE, BIA, body composition, phase angle, leanness

## Abstract

Background: The prevalence of anorexia nervosa among males is increasing but few data are available in the literature. This cross sectional study aims to evaluate resting energy expenditure (REE) and phase angle as a marker of qualitative changes of fat free mass (FFM) in three leanness groups as compared with control subjects. Methods: 17 anorectic (AN) males, 15 constitutionally lean (CL) individuals, 12 ballet dancers (DC), and 18 control (CTR) subjects were evaluated. REE was measured by indirect calorimetry (V max29- Sensormedics), and body composition was evaluated by bioimpedance analysis (BIA) at 50 kHz (DS Medica). Phase angle (a bioimpedance variable related to nutritional status) was used to evaluate differences in FFM characteristics between these three types of leanness. Results: REE, adjusted for FFM and fat mass (FM), were significantly higher in CL and lower in AN individuals (1783 ± 47 vs. 1291 ± 58 kcal, *p* < 0.05) compared to the other groups. Body composition was similar in AN and CL whereas dancers had the highest FFM (58.9 ± 4.8 kg, *p* < 0.05); anorectic males showed the lowest phase angle (5.8 ± 1.2 degrees vs. other groups, *p* < 0.05) and dancers the highest phase angle (7.9 ± 0.7 degree vs. other group, *p* < 0.05). Conclusions: Our findings confirm that phase angle could be a useful marker of qualitative changes, above all in the field of sport activities. On the other hand, there is the need to further evaluate the relationship between resting energy expenditure, body composition and endocrine status in different conditions of physical activity and dietary intake.

## 1. Introduction

The World Health Organization has suggested three underweight categories: grade 1 (mild underweight; body mass index (BMI): 17–18.5 kg/m^2^), grade 2 (moderate underweight; BMI: 16–16.9 kg/m^2^), and grade 3 (severe underweight; BMI: <16 kg/m^2^) [1].

The most common underweight condition in westernized young females is restrictive anorexia nervosa (AN); its prevalence is reportedly between 1–4% of young women and 0.3–0.7% of young males in Europe [2,3]. Although considerable research into body composition and its effect on several clinical outcomes has been conducted in female anorectic patients, few data are available in anorectic males, despite a slow but continuous increase in prevalence, at least in current clinical practice [4,5]. 

To our knowledge, relatively few studies have focused on characteristics of AN males [6,7]. Some studies in adolescent males with AN have shown that FFM is an important determinant of bone mineral density [8,9], as is the case in anorectic females, and there is a greater risk of osteoporosis in male patients [10]; moreover, further research is warranted in male anorectic populations due to their different endocrine patterns.

On the other hand, underweight is a condition that could be observed both in female and male adolescents with constitutional leanness or in the field of sport activities, such as professional ballet dancers, where leanness is a basic requirement [11].

Constitutional leanness (CL) is generally defined as thinness not due to organic disease or anorexia nervosa. In these subjects, body mass index (BMI) is always in the lower percentiles for age and gender, without any hormonal or other pathological abnormalities [12]. Nowadays, the characteristics of lifelong thin individuals [13] are not well known and further evidence is needed to identify which criteria should be used to characterize a constitutionally thin male [14]. 

A recent study showed that resting energy expenditure was increased in children with constitutional leanness, and this increased energy expenditure-to-FFM ratio differentiated CL from controls [15]. To the best of our knowledge, only a few studies have been carried out on energy regulation in CL. In our experience [16], we found that brown adipose tissue (BAT) could have a role in regulatory (cold- or diet-induced, etc.) thermogenesis and that REE is higher in CL than in AN women, even in refeed AN; however BAT activity is well represented in resting thermoneutral conditions in CL individuals and is correlated with resting energy expenditure (REE), particularly when corrected for FFM. Thus, the results of the study suggested that having active BAT contributed to constitutional leanness. 

In sports medicine, there are few studies on REE and body composition in professional dancers. A previous study from our group has shown that in young ballet dancers, a minimum weight is required in order to adequately perform physical activity [11]. Recently, Ferrari et al. [17] studied 13 professional ballet dancers of the Bolshoi theatre company and 22 university physical education students (11 females and 11 males), and the researchers found that the intense training in classical ballet significantly modified body composition components.

To our knowledge, no other studies have compared body composition in these forms of leanness. In particular, phase angle is considered a good nutritional marker, as it is related to the ratio of intra- to extracellular water and body cell mass [18]. 

Therefore, the aim of this study was to evaluate body composition and phase angle, among BIA parameters, REE, both in absolute values and corrected and adjusted for FFM, in these types of leanness (anoressia nervosa, constitutional leanness and ballet dancers). 

## 2. Subjects and Methods

### 2.1. Patients

Sixty-two young males evaluated at the Clinical Nutrition Unit of the Department of Clinical Medicine and Surgery, Federico II University Hospital, Naples, were enrolled in this cross sectional study: 17 clinically stable anorectic young men (with a stable weight from at least six months according to DSM-V, 2013) [19], 15 constitutionally lean individuals, 12 ballet dancers from a classical school and 18 control subjects. 

All patients and controls underwent routine laboratory evaluations and bioimpedance analysis. 

There were no additional selection criteria for constitutionally lean males other than the absence of illness. Anorectic patients were recruited from our Clinical Nutrition Unit and studied at entry before any nutritional treatment; all subjects were on their usual diet without any dietetic recommendation. Constitutional leanness patients were referred to our Clinical Nutrition Unit for a medical evaluation of their condition of underweight, stable on time; after an in depth clinical evaluation, in absence of any pathology, they are classified as CL. The control group is representative of our local university students and they don’t perform sport activity regularly, only leasure time activity.

The study was carried out according to the Declaration of Helsinki, and the protocol was approved by the local ethics committee at Federico II University Hospital. All subjects, or their parents when required, gave informed consent for all diagnostic evaluations. All measurements were taken under fasting conditions early in the morning.

### 2.2. Anthropometry

Weight was measured to the nearest 0.1 kg using a platform beam scale, and height was measured to the nearest 0.5 cm using a stadiometer (Seca 709; Seca, Hamburg, Germany). BMI was determined according to the standard formula of body weight measured in kilograms divided by height in metres squared.

### 2.3. Bioimpedance Analysis (BIA)

Bioimpedance analysis (BIA) was performed at 50 kHz using Human Im Plus II (DS Medica-Milan) at room temperature (22–25 °C) after 20 min in the supine position; all subjects were in a fasted state (12 h) and voided prior to measurement. FFM and fat mass (FM) were estimated using the prediction equations developed by Kushner [20].

Each subject was evaluated on the non-dominant side of the body using disposable electrodes. Two injecting electrodes were placed on the hand and foot (mid-dorsum of hand/foot just proximal to the metacarpal/metatarsal phalangeal joint line) and two sensing electrodes were placed on the wrist and ankle (mid-dorsum of wrist/ankle centred on a line joining the bony prominences of radius and ulna/the medial and lateral malleoli).

The measured BIA variables were resistance (R), reactance (Xc) and phase angle (PhA). Bioimpedance index (BI-index in cm^2^/ohm) was derived as the ratio height^2^/whole-body R [21].

### 2.4. Resting Energy Expenditure

Resting energy expenditure (REE) and respiratory quotient (RQ) was measured by indirect calorimetry using a canopy system (V max29, Sensor Medics, Anaheim, USA.) at room temperature (23–25 °C). The instrument was checked by burning ethanol, while oxygen and carbon dioxide analysers were calibrated using standardized gases (nitrogen, carbon dioxide and oxygen mixture). All patients were in the post-absorptive condition (12–14 h fasting) and in a quiet environment on the bed. After a 15-minute adaptation period, oxygen consumption and carbon dioxide production were determined for 45 minutes. Energy expenditure was calculated according to the Weir’s formula, neglecting protein oxidation [22].

### 2.5. Statistical Analysis

All data were computerized and analysed with the software SPSS, WIN version 18 (SPSS, Chicago, Illinois, USA). All results are expressed as the mean and SD. One-way analysis of variance (ANOVA) with post-hoc tests was performed to compare data between different groups. *p* values < 0.05 were considered statistically significant.

## 3. Results

Table 1 shows anthropometric parameters of the four groups. Age and height were not significantly different between groups; weight and BMI were highest in control subjects and similar in anorectic and lean males, in particular they were higher in dancers and controls males than in anorectic and constitutionally lean subjects.

In terms of body composition (Table 2), FFM was similar in AN and CL males, and these values were significantly lower than in dancers and controls. Absolute FM and in percentage was significantly lower in dancers and higher in controls; there were no differences between anorectic males and constitutionally lean individuals.

CL and CTR had a similar PhA, whereas anorectic males had the lowest PhA (5.8 ± 1.2 degrees) and dancers had the highest PhA (7.9 ± 0.7 degrees) (Table 2).

REE was significantly lowest in AN individuals compared to constitutionally lean individuals, dancers and controls (Table 3). Respiratory quotient (RQ) results were similar among the four groups (Table 3). The results of REE adjusted for FFM did not differ from the measured REE (except for dancers −65 kcal/day) (Table 3), whereas REE corrected per FFM was significantly highest in CL and there were no differences between AN and dancers (Table 3). As expected, we found a significant linear correlation between REE and FFM (R^2^ = 0.195) but interestingly there were also a significant linear correlation between REE and FM (R^2^ = 0.171).

## 4. Discussion

This study shows that there are differences in body composition, in particular the percentage of body fat was similar between constitutionally lean males and anorectic patients with similar BMIs, whereas dancers show the lowest amount of body fat and higher amounts of FFM compared to controls, despite differences in body weight and BMI.

In a previous study of females [11], it was shown that there were no differences in anthropometric and bioelectrical impedance analysis (BIA) variables between three groups of underweight young females: anorectic, constitutionally lean, ballet dancers, compared to control subjects. An explanation for these results could be related to the different type of physical activity performed by males and females: training in young women focuses on the lower limbs, whereas males also strengthen upper body muscles. 

Several studies have shown that phase angle is a parameter that can be used to monitor disease progression [23]. From a nutritional viewpoint, phase angle decreases in different forms of malnutrition and is also a predictor of survival in several pathological conditions [17,24]. As already observed in young females with anorexia nervosa [25,26], chronic underfeeding contributes to a decrease in phase angle, which is most likely due to an increase of extracellular water and/or a decrease in body cell mass.

In our view, an original finding of this study is that phase angle, a BIA-derived parameter, was confirmed as a marker of nutritional status that could discriminate types of underweight: it resulted significantly highest in dancers (physical activity), lowest in anorectic patients (diseases) and similar between constitutionally lean (physiological) patients and controls.

Regarding resting energy expenditure, many studies have focused on underweight AN female patients [13,27] because the assessment of REE plays an important role in the dietary management of this pathological condition. 

In our study, absolute REE and REE adjusted for FFM were both significantly lower in the AN group and higher in the CL group than in control subjects; however, it seems more useful to express REE in terms of FFM rather than in absolute values; thus, REE corrected for kg of FFM was greater in CL individuals than in controls. 

In constitutional thinness, several studies reported a significantly higher REE when corrected for kg of FFM than in control subjects [12,13,15] This increased energy expenditure-to-FFM ratio distinguishes CL individuals from controls and could account for the resistance to weight gain observed in the CL condition [12,16,28]. These metabolic differences are probably genetically determined and influenced by hormonal levels [29,30].

Unexpectedly, dancers showed REE values similar to the anorectic group, also after adjustment for FFM. In the literature there are few studies on the relationship between REE and physical activity, at least in these physiological and clinical conditions [31]. A possible reason for this finding is the drastically low fat mass, which brings about a metabolic adaptation in form of a down-regulation of REE, possibly mediated by leptin as shown in many previous studies on REE in female patients with anorexia nervosa [32]; another possible reason could be the metabolic adaptation at the physical activity, considering that professional ballet dancers have biomechanical changes and functional performance related to intense dance training [33].

The limitation of this study is that data on leptin or thyroid status were not available, so we cannot establish a possible relationship with endocrine data; however, this is a preliminary cross sectional study with a small sample size due to the uncommon condition of the subjects. So there is the need to compare our data with a multicenter study. 

In conclusion, in the clinical practice the measurements of phase angle may be useful to discriminate between these three different forms of leanness in males (physical activity, disease and physiological condition) as well whereas the evaluation of REE is important for the nutritional approach of the patients. However, our findings highlight the need to further evaluate the relationship between resting energy expenditure and physical activity and dietary intake.

## Figures and Tables

**Table 1 nutrients-11-00502-t001:** Anthropometric measurements of 62 males individuals, according to leanness groups.

	AN (*n* = 17)	CL (*n* = 15)	DANCERS (*n* = 12)	CTR (*n* = 18)
Age (years)	22.3 ± 5.3	23.3 ± 5.2	19.7 ± 1.6	22.3 ± 3.7
Weight (kg)	51.8 ± 4.8 ^a^	56.1 ± 3.3 ^a^	62.3 ± 5.3 ^b^	70.3 ± 6.5
Height (cm)	174 ± 5.1	177 ± 4.5	176 ± 5.0	177 ± 4.4
BMI (kg/m^2^)	17.1 ± 1.2 ^a^	17.9 ± 0.6 ^a^	20.0 ± 1.3 ^b^	22.3 ± 1.7

AN = anorectic; CL= constitutional leanness; CTR= control; ^a^
*p* < 0.05 vs. Dancers and CTR; ^b^
*p* < 0.05 vs. CTR.

**Table 2 nutrients-11-00502-t002:** Body composition of 62 males individuals, according to leanness groups.

	AN (*n* = 17)	CL (*n* = 15)	DANCERS (*n* = 12)	CTR (*n* = 18)
FFM (kg)	46.0 ± 5.2 ^a^	48.6 ± 4.4 ^a^	58.9 ± 4.8	56.2 ± 6.1
FM (kg)	5.3 ± 2.8 ^b^	7.6 ± 3.1 ^a^	3.4 ± 1.3 ^b^	14.0 ± 30
FM (%)	10.3 ± 5.7 ^a^	13.6 ± 5.7 ^a^	5.5 ± 1.8 ^c^	20.0 ± 3.6
PhA (degrees)	5.8 ± 1.2 ^c^	6.9 ± 0.6 ^d^	7.9 ± 0.7 ^b^	6.8 ± 0.4

^a^*p* < 0.05 vs. Dancers and CTR; ^b^
*p* < 0.05 vs. CTR; ^c^
*p* < 0.05 vs. AN and Dancers and CTR; ^d^
*p* < 0.05 vs. Dancers, FFM: fat Free Mass, FM: Fat Mass, PhA: phase angle.

**Table 3 nutrients-11-00502-t003:** Resting energy expenditure of 62 males individuals, according to leanness groups.

	AN (*n* = 17)	CL (*n* = 15)	DANCERS (*n* = 12)	CTR (*n* = 18)
REE (kcal/day)	1150 ± 169 ^a^	1726 ± 216 ^b^	1563 ± 179 ^c^	1678 ± 167
REE adjusted for FFM (kcal/day)	1185 ± 57 ^a^	1757 ± 48 ^b^	1498 ± 60 ^a^	1637 ± 46 ^b^
REE/FFM (kcal/kg)	25.2 ± 4.2 ^d^	35.9 ± 6.2 ^a^	26.6 ± 2.4 ^e^	30.0 ± 3.0 ^f^
RQ	0.84 ± 0.06	0.85 ± 0.05	0.86 ± 0.07	0.85 ± 0.04

^a^*p* < 0.05 vs. all groups; ^b^
*p* < 0.05 vs. AN and Dancers; ^c^
*p* < 0.05 vs. AN; ^d^
*p* < 0.05 vs. CL and CTR; ^e^
*p* < 0.05 vs. CL; ^f^
*p* < 0.05 vs. AN and CL. REE: resting energy expenditure, RQ: respiratory quotient.

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
