# Peer review of "Resting Energy Expenditure, Body Composition and Phase Angle in Anorectic, Ballet Dancers and Constitutionally Lean Males"

_nutrients, 2019, doi:10.3390/nu11030502_

Round 1

Reviewer 1 Report

An interesting and novel topic relevant to the field.  A few comments/suggestions:

Be consistent in using either REE or. RMR throughout the paper (use one or the other). 

It is unclear why 18 'controls' were used, and how this group was defined.  Would clarify the significance of using these 4 groups in your design- vs. using 3 groups:  anorectic males, constitutionally lean males, and male ballet dancers.  Also note a BMI of 20 and 22 (Dancers and Controls) are considered "healthy weight" range vs. BMI 17  "Underweight". (Your AN and CL groups).  From this perspective, some of your data are not surprising or novel.    

Using 3 groups seems more consistent with your second Aim (Lines 75-77)

Would describe how you selected your constitutionally lean males (methods)

Would describe standards for measuring BIA.  For example, being in a fasted state could impact your results. 

Please describe your methods in more detail with regard to number of measurements and over what time frame.  Did you measure BIA, REE and phase angle only once for each research participant? If so, this needs to be noted as a limitation of your study. How long did the study last? 

Lines 118-119.  Would confirm that the inter-day variation that you used is appropriate (i.e. six obese individuals).  Why were current research participants not used? 

The most relevant and interesting data seems related to AN and CL, would focus discussion and conclusions on these points.  Would be clear how your data support your 2 aims (Lines 72-77). 

Would discuss how REE and REE/FFM compares in AN and CL- these points seem  relevant to your Aim 2, and to the use of phase angle.  This is not clearly discussed. 

would discuss the significance of the similar REE/FFM between AN and Dancers- yet had the most significant difference in phase angles.(lines 135-136)

Lines 194-195:  Interesting conclusion, and clinically relevant; however your discussion of your data needs to support this conclusion more clearly. 

also recommend revising the sentence (lines 194-195): are you trying to say:  "In conclusion, the measurement of phase angle may be useful in clinical practice to discriminate among forms of underweight in male athletes,  such as anorectic or constitutionally lean"? 

Author Response

Reviewer 1

An interesting and novel topic relevant to the field.  A few comments/suggestions:

Be consistent in using either REE or. RMR throughout the paper (use one or the other). 

We use REE throughout the paper (line 60)

It is unclear why 18 'controls' were used, and how this group was defined.  Would clarify the significance of using these 4 groups in your design- vs. using 3 groups:  anorectic males, constitutionally lean males, and male ballet dancers.  Also note a BMI of 20 and 22 (Dancers and Controls) are considered "healthy weight" range vs. BMI 17  "Underweight". (Your AN and CL groups).  From this perspective, some of your data are not surprising or novel.    

First of all, we would like to clarify that our control group has a ratio 1:2 compared with the total research study group; they are representative of our local university students and they don’t perform sport activity regularly, only leasure time activity. (we add this definition in the text, line 89-91).

So, the aim of our study was to compare these three different leanness (AN, dancers e CL) both for body composition and REE because despite the group of dancers was considered “healthy weight”, it included also underweight subjects (mean BMI 20.0 ±1.3 Kg/m²); furthermore, in order to be more rigorous, we modified the definition of “underweight group” in “leanness group” throughout the paper .

Using 3 groups seems more consistent with your second Aim (Lines 75-77)

We have modified the text with only one aim (line 72-74)

Would describe how you selected your constitutionally lean males (methods)

We have described this condition in the methods section (line 87-89)

Would describe standards for measuring BIA.  For example, being in a fasted state could impact your results. 

We have described the standards for measuring BIA at line 103-104

Please describe your methods in more detail with regard to number of measurements and over what time frame.  Did you measure BIA, REE and phase angle only once for each research participant? If so, this needs to be noted as a limitation of your study. How long did the study last? 

We confirm that this a retrospective, cross sectional study so all evaluations  were carried out only once. This could be a limitation of our study, but we have clarified these point in the discussion (line 197-199)

Lines 118-119.  Would confirm that the inter-day variation that you used is appropriate (i.e. six obese individuals).  Why were current research participants not used? 

Yes we confirm that the inter-day variation used is appropriate because usually the subjects used for the standardization are underweight, normal weight and obese.

The most relevant and interesting data seems related to AN and CL, would focus discussion and conclusions on these points.  Would be clear how your data support your 2 aims (Lines 72-77).

We have modified the discussion according to your suggest

Would discuss how REE and REE/FFM compares in AN and CL- these points seem  relevant to your Aim 2, and to the use of phase angle.  This is not clearly discussed. 

We modified the discussion (lines 179-180) and we have added a new parameter in the table 3 (REE adjusted for FFM) in order to better explain the ratio between REE and FFM in all groups.

would discuss the significance of the similar REE/FFM between AN and Dancers- yet had the most significant difference in phase angles.(lines 135-136)

we modified the discussion and we have added a new reference in order to better explain the difference between AN and Dancers (ref n 34, lines 193-195)

Lines 194-195:  Interesting conclusion, and clinically relevant; however your discussion of your data needs to support this conclusion more clearly. 

We modify the discussion according to your suggest

also recommend revising the sentence (lines 194-195): are you trying to say:  "In conclusion, the measurement of phase angle may be useful in clinical practice to discriminate among forms of underweight in male athletes,  such as anorectic or constitutionally lean"? 

 We modified the sentence (lines 200- 202)

Reviewer 2 Report

This is an interesting study in under-researched populations. It confirms the expected body composition values in these different states and adds new data on phase angle. 

Major comments

It is unclear to me why the authors chose a two-way ANOVA as their preferred model, given that a one-way ANOVA with post-hoc tests would be the most suitable one for their comparisons (or an equivalent non-parametric test). Can you please expand on this choice or revise as appropriate? 

Minor comments

Please mention your study design in the title or abstract

line 26: typo, "confirms"

lines 52-3: "very common condition" is unclear, please rephrase to make it more specific, how common?

lines 73, 81, 187. 195: delete extra spaces 

line 131: reword "as far as" with "in terms of "

Methods

Please expand on your study design, it is unclear to me how this is a retrospective study (e.g. was it a chart review?), my initial thought when reading only the abstract was that this is a cross-sectional one. 

Results

lines 137-8. Please check the accuracy of this sentence. It reads as if you are comparing CL to CL.

Discussion

line 182: No need to start a new paragraph. Overall, please avoid one-sentence paragraphs. 

line 190: Change "limit" to "limitation"

Limitations of the study: Please mention that the data should be regarded as preliminary and the small sample size and low power to detect differences should be acknowledged unless sample size was calculated a priori and should be reported. 

line 184: The references do not support this sentence, please consider revising. 

line 195-6: It is unclear what the authors mean by "the knowledge of REE is important for the nutritional approach of the patients"

Please add a completed STROBE statement as an appendix to the study. 

Table 3: change kcal/die to kcal/day

Author Response

Reviewer 2

This is an interesting study in under-researched populations. It confirms the expected body composition values in these different states and adds new data on phase angle. 

Major comments

It is unclear to me why the authors chose a two-way ANOVA as their preferred model, given that a one-way ANOVA with post-hoc tests would be the most suitable one for their comparisons (or an equivalent non-parametric test). Can you please expand on this choice or revise as appropriate? 

We apologize for the error, we have revised the text. (line 125-126)

Minor comments

Please mention your study design in the title or abstract

line 26: typo, "confirms"

Corrected

lines 52-3: "very common condition" is unclear, please rephrase to make it more specific, how common?

We have modified the text as suggested (line 52-53)

lines 73, 81, 187. 195: delete extra spaces 

Done as suggested

line 131: reword "as far as" with "in terms of "

Done as suggested

Methods

Please expand on your study design, it is unclear to me how this is a retrospective study (e.g. was it a chart review?), my initial thought when reading only the abstract was that this is a cross-sectional one. 

We confirm that this a retrospective, cross-sectional study

Results

lines 137-8. Please check the accuracy of this sentence. It reads as if you are comparing CL to CL.

We have checked the accuracy of the sentence (Lines 139-140)

Discussion

line 182: No need to start a new paragraph. Overall, please avoid one-sentence paragraphs. 

Corrected as requested.

line 190: Change "limit" to "limitation"

Done as suggested

Limitations of the study: Please mention that the data should be regarded as preliminary and the small sample size and low power to detect differences should be acknowledged unless sample size was calculated a priori and should be reported

We have added this limitation in the discussion (line 197-198)

line 184: The references do not support this sentence, please consider revising. 

We modified the text in order to justify the references (line 187)

line 195-6: It is unclear what the authors mean by "the knowledge of REE is important for the nutritional approach of the patients"

we have revised the sentence (line 202)

Please add a completed STROBE statement as an appendix to the study. 

We have added the STROBE statement as an appendix (pages 8-9-10)

Table 3: change kcal/die to kcal/day

Corrected

Round 2

Reviewer 1 Report

A much improved presentation of the study design, and discussion of results and conclusions

Suggestions for further improvement:

 Lines 59-62 starting with "we also found that BAT..."  does not add to the purpose of this particular study. Recommend taking these lines out.

Would reword line 87 to state something like"  Constitutional lean patients were referred to...." vs. current wording "Constitutional leanness are patients.."

Please take out lines 120-121 re: Inter day coefficient variation..  This measurement should be performed using the study population and as relevant to the particular study.

In Table 3,  pleas add "s" to  adjusted in "REE adjusted for FFM"

Would reword line 171 to " In our view, an original finding of this study is that phase angle...
(take out 'leanness males")

Line 179 would reword.  Suggestion:  "In our study, absolute REE and REE adjusted for FFM were both significantly lower..."

Another recommendation for future study:  while BIA is clinically meaningful, and can be a relatively valid measurement of body composition, it will be important to validate your body composition results with the use of a more valid measure of body composition such as DEXA or air displacement plethysmograpy (i.e. BodPod)

Author Response

A much improved presentation of the study design, and discussion of results and conclusions

Suggestions for further improvement:

 Lines 59-62 starting with "we also found that BAT..."  does not add to the purpose of this particular study. Recommend taking these lines out.

Removed as suggested (line 60)

Would reword line 87 to state something like"  Constitutional lean patients were referred to...." vs. current wording "Constitutional leanness are patients.."

Done as suggested

Please take out lines 120-121 re: Inter day coefficient variation..  This measurement should be performed using the study population and as relevant to the particular study.

Removed suggested

In Table 3,  pleas add "s" to  adjusted in "REE adjusted for FFM"

Done as suggested

Would reword line 171 to " In our view, an original finding of this study is that phase angle...
(take out 'leanness males")

Done as suggested

Line 179 would reword.  Suggestion:  "In our study, absolute REE and REE adjusted for FFM were both significantly lower..."

Done as suggested

Another recommendation for future study:  while BIA is clinically meaningful, and can be a relatively valid measurement of body composition, it will be important to validate your body composition results with the use of a more valid measure of body composition such as DEXA or air displacement plethysmograpy (i.e. BodPod)

Thanks for your suggest. It’s a very strong recommendation for future study.

Reviewer 2 Report

The authors have mostly addressed my concerns satisfactorily. 

The only remaining issue is the study design. I suggest that the authors reword "retrospective cross-sectional" to "cross sectional" throughout the paper AND mention the study design in the abstract or title.  

Author Response

The authors have mostly addressed my concerns satisfactorily. 

The only remaining issue is the study design. I suggest that the authors reword "retrospective cross-sectional" to "cross sectional" throughout the paper AND mention the study design in the abstract or title.  

We modified the text as requested and we have added the definition of cross sectional study in the abstract.(line 14, 79, 197)
